# Host Factors Affect the Gut Microbiome More Significantly than Diet Shift

**DOI:** 10.3390/microorganisms9122520

**Published:** 2021-12-06

**Authors:** Enkhchimeg Lkhagva, Hea-Jong Chung, Ji-Seon Ahn, Seong-Tshool Hong

**Affiliations:** 1Department of Biomedical Sciences and Institute for Medical Science, Chonbuk National University Medical School, Jeonju 54907, Korea; enkhchmg2580@gmail.com; 2Gwangju Center, Korea Basic Science Institute, Gwangju 61715, Korea; hjchung84@kbsi.re.kr (H.-J.C.); ajs0105@kbsi.re.kr (J.-S.A.)

**Keywords:** diet shift, exercise, gut microbiome, host factors, the composition of gut microbiome

## Abstract

The determining factors of the composition of the gut microbiome are one of the main interests in current science. In this work, we compared the effect of diet shift (DS) from heavily relying on meatatarian diets to vegetarian diets and physical exercise (EX) on the composition of the gut microbiome after 3 months. Although both DS and EX affected the composition of the gut microbiome, the patterns of alteration were different. The α-diversity analyzed by InvSimpson, Shannon, Simpson, and Evenness showed that both EX and DS affected the microbiome, causing it to become more diverse, but EX affected the gut microbiome more significantly than DS. The β-diversity analyses indicated that EX and DS modified the gut microbiome in two different directions. Co-occurrence network analysis confirmed that both EX and DS modified the gut microbiome in different directions, although EX modified the gut microbiome more significantly. Most notably, the abundance of *Dialister succinatiphilus* was upregulated by EX, and the abundances of *Bacteroides fragilis*, *Phascolarctobacterium faecium*, and *Megasphaera elsdenii* were downregulated by both EX and DS. Overall, EX modulated the composition of the gut microbiome more significantly than DS, meaning that host factors are more important in determining the gut microbiome than diets. This work also provides a new theoretical basis for why physical exercise is more health-beneficial than vegetarian diets.

## 1. Introduction

Living organisms in nature exist as communities of various species, interacting closely with each other. Microbial organisms are also present as heterogeneous populations. Microbial organisms frequently appear as a dense mixture of various species to interact with each other in nature. Considering the heterogeneous presence of microbial organisms, it would be reasonable to ponder that the emergence of the first primitive multicellular organism could be accompanied by the gut microbiome at its beginning. Therefore, the fundamentals of host–gut microbiome interactions and their evolutionary consequences would open a new horizon for understanding animals [1,2].

The gut microbiome has been coevolving with humans throughout its evolutionary history [3,4,5,6]. Recent studies have shown that the gut microbiome plays significant determinant roles in almost all phenotypes of animals, including diseases, as much as the genomes of their hosts [5,6,7,8,9]. The stability and dynamics of the gut microbiome have not only local but also systemic effects that determine the phenotypes and diseases of the host [10]. As the genes of an animal are a result of eons of natural selection, the gut microbiome of an organism is also the result of long natural selection to modulate the phenotypes of its host. Recent works suggest that the gut microbiome is a surprising factor that determines the phenotypes of mammals, similar to their own genes [5,6,7,8,9].

Given the intimate and complex interactions between the gut microbiome and its host during evolutionary history, host factors could dictate the composition of the gut microbiome. However, despite a possible significant role of host factors in determining the composition of the gut microbiome, the contemporary prevailing opinion considers diet as the main determinant factor for the composition of the gut microbiome [5,6,11]. It has been well documented that dietary shifts affect the composition of the gut microbiome [11,12,13,14]. The opinion that the composition of the gut microbiome is mainly determined by diet was supported by research showing that the gut microbiomes of animals with similar dietary niches tend to contain similar intestinal microbes [15,16]. Although the effect of diet on the composition of the gut microbiome is clear, recent research has suggested that host factors may also play a role in determining the composition of the gut microbiome [17,18,19]. These results collectively propose that the composition of the gut microbiome is determined by both diet and host factors. However, it is unclear which factor plays a more important role.

Contrary to expectations, a recent study showed that the number of dietary transitions within an evolutionary lineage did not influence rates of microbiome divergence, but, instead, the most dramatic changes in the gut microbiome were associated with the physiological changes of the species during the evolutionary process [20]. This work strongly suggests that host factors could impact the composition of the gut microbiome as much as diet or even more. The effect of host factors on the composition of the gut microbiome was further validated by a recent report that the change in host physiology during evolutionary processes outweighs dietary change in structuring the gut microbiomes of primates [21]. Based on these two works suggesting that the change of host factors affects the composition of the gut microbiome more than dietary change during the emergence of a species during evolution, the significance of host factors in a human individual would be a very interesting question. However, the significance of host factors affecting the gut microbiome has not been investigated.

Considering such significant roles of the gut microbiome in humans from early evolutionary history to the present, it would be much more favored in natural selection if humans were able to determine their own gut microbiome. Therefore, the significance of host factors in determining the human gut microbiome would be an important question to answer with respect to human biology. In this work, we conducted a comparative study to investigate which one is more relevant in determining the diversity of the gut microbiome between host factors and diet shift.

## 2. Materials and Methods

### 2.1. Study Design

A 12-week, randomized, parallel, controlled clinical trial was carried out with diet interventions at the Clinical Trial Center for Functional Foods (CTCF2) in the Chonbuk National University Hospital, South Korea. We recruited 30~50-year-old volunteers depending on a meat-containing diet at least twice per day to investigate the compositional change in the gut microbiome after a diet shift to a vegetarian diet or a physiological shift by exercise. Computer-generated random numbers were used to assign each subject to either the experimental or control group. The 75 volunteers were divided into three groups: one group shifting their diet from a meat diet to a vegetarian diet (the DS group), the second group adopting a 30 min physical exercise regimen of a guided aerobic exercise in a fitness center three times per week without changing their original diets (the EX group), and the control subjects continuing their lifestyle (the Ctrl group). After 3 months, the volunteers were interviewed to ask whether they strictly followed the experimental guidelines, and fecal samples from 41 individuals who followed the guidelines were collected for further analysis (DS group, *n* = 14; EX group, *n* = 13; Ctrl group *n* = 14) (see also Appendix A).

### 2.2. Fecal Sample Collection and DNA Preparation

Fecal samples were freshly collected 2 times from each participant at the beginning of the study (week 0) and at the end of the intervention (week 12). Fecal samples were kept in individual sterile feces containers at 4 °C and processed within 4 h. Each sample was mixed in an equal volume of sterile phosphate-buffered saline buffer and homogenized using a stomacher machine before aliquoting. Aliquots of 1 mL were frozen immediately at −80 °C for further processing. The fecal samples collected from four random sites from each individual feces were mixed together before genomic DNA isolation. Genomic DNA was extracted from ~1 g fecal aliquot sample using the Mobio PowerLyzer™ PowerSoil^®^ DNA Isolation Kit (Qiagen, Hilden, Germany). The DNA extraction procedure followed the standard protocol supplied by the company, and the final elution of DNA was performed with 100 μL Tris (MoBIO buffer C6). The quantity and quality of the purified genomic DNA were evaluated by an absorbance spectrophotometric method using a BioSpec-nano spectrophotometer (Shimadzu, Kyoto, Japan), and the purified DNAs were stored at −20 °C until sequencing.

### 2.3. Microbial Genomic Sequencing and Data Analysis

Metagenome sequencing analyses of the gut microbiome DNA samples were processed and sequenced by a commercial company, Chunlab, Inc. in South Korea. Amplification of genomic DNA was performed using barcoded primers targeting the V1 to V3 regions of the bacterial 16S rRNA gene (V1-9F: 5′-X-AC-GAGTTTGATCMTGGCTCAG-3′ and V3-541R: 5′-X-AC-WTTACCGCGGCTGCTGG-3′, where X is a unique barcode for each sample, followed by a common linker, AC). The amplified DNA was then sequenced using a 454 GS FLX Titanium Sequencing System (Roche, Bradford, CT, USA). Sequencing reads of each sample were separated by unique barcodes. After sequencing, the sequences of barcode, linker, and PCR primer at both sides were removed from the original sequencing reads. Only reads containing 0–1 ambiguous base calls (Ns) and 300 or more base pairs were selected for the final bioinformatic analyses from the resultant sequences. Non-specific PCR amplicons that showed no match with the 16S rRNA gene database upon BLASTN search (expectation value of >10^−5^) were also discarded.

The sequence reads (see also Appendix A) generated from metagenome sequencing were identified using the EzTaxon-e database (http://eztaxon-e.ezbiocloud.net/ Accessed on 10 July 2021) [22,23,24]. Mothur, an open-source bioinformatics pipeline, was used to analyze sequences to assign operational taxonomic units (OTUs) and generate taxonomy classification [25]. A cutoff value of 97% similarity of the 16S rRNA gene sequences was defined as the same species. The raw data were deposited in the repository at figshare (https://doi.org/10.6084/m9.figshare.16620349.v1).

### 2.4. Data Normalization and Differential Abundance Analysis

The DESeq2 package was used to identify the bacteria with the most significant changes in differential abundance at the species level in each sample. The raw read count data were processed based on the median of ratio normalization method using the DESeq2 package within the R program. The counts were divided by sample-specific size factors determined by the median ratio of species counts relative to geometric mean per species. All normalized counts were exported as an Excel table and used for further analysis. Differential abundance was identified by Wald test in the DESeq2 package by using three pairs of group comparison: (1) EX to Ctrl, (2) DS to Ctrl, (3) EX to DS. The filter criterion was an adjusted *p* value < 0.05. 

### 2.5. α-Diversity and Abundance Evaluation of Microbiome

We used the phyloseq (1.28.0) [26] and metagenomeSeq (1.16.0) [27] packages to identify the central taxa present in each group. The metadata, OTUs, and taxonomic classification tables were imported into the phyloseq package and the data were processed as instructed [28,29]. The phyloseq class object was converted to metagenomeseq objects and normalized by cumulative-sum-scaling (CSS), which was specially built for metagenome data in the bioConductor package metagenomeSeq (1.16.0) [27]. Normalized data were converted to phyloseq class objects in R for further analysis and visualization.

Normalized OTU data were used for abundance calculation, and each taxonomic level was glommed for plotting. For clear visualization of abundance data, taxa were collected into “other” if they had relative abundances below 5%, except at the phylum and class levels (Appendix A).

### 2.6. β-Diversity and Abundance Evaluation of Microbiome

β-diversity metrics were computed and visualized using log-transformed, normalized OTU data in the phyloseq package using Bray–Curtis dissimilarity. The unweighted UniFrac metric was used for β-diversity and PCoA was calculated and visualized by the vegan package [30], while NMDS was plotted in the phyloseq package in R. The significances of β-diversity metrics were tested by analysis of dissimilarity (ADONIS) with 999 permutations by the vegan package [30].

### 2.7. Construction of Heatmap and Phylogenetic Tree

A heatmap and cluster analysis were generated using the relative abundances of genera from all OTU values or core abundant OTU values in the Heatplus (2.30.0) package from bioconductor and the vegan package in R. Average linkage hierarchical clustering and Bray–Curtis distance metrics were used for cluster analysis and heatmap generation, respectively [31]. Unsupervised prevalence filtering was performed with a 5% threshold in total samples to collect the most abundant taxa for heatmap generation.

Phylogenetic trees for each sampling site were constructed from row sequences without any filtering to show direct visualization of sample richness with relation to taxonomic classification. Taxa that could not be classified down to the species level were reclassified based on the NCBI accession number using the taxonomizr (0.5.3) package in R [32]. Then, 16S rRNA sequences from each sampling site were aligned in ClustalW [33] with a default parameter, and the resulting alignments were used to construct maximum-likelihood phylogenetic trees in MEGAX [34] with 500 bootstrap replicates. All phylogenetic trees were visualized in iTOL [35].

### 2.8. Co-Occurrence Network Construction

Co-abundance networks were created by the ReBoot20 algorithm [36], known as a permutation–renormalization–bootstrap network construction strategy, to study how diet shift and exercise affect microbial co-occurrence relationships. Non-normalized abundance data were uploaded to CoNet [37], a Java Cytoscape plug-in. Tree networks were independently constructed by splitting the OTU abundance matrix into Ctrl, EX, DS groups. The microbial networks and links or edges were obtained from OTU occurrence data. The multiple ensemble correlation method in CoNet was used to identify significant copresence across the samples, while OTUs that occurred in less than three samples were discarded (“row_minocc” = 3). Five similarity measures, including Spearman and Pearson correlation coefficients, the Mutual Information Score, and the Bray–Curtis and Kullback–Leibler Dissimilarity, were calculated by CoNet for the creation of an ensemble network and the *p* value was merged by Brown’s method. The *p* value was corrected by the Benjamini–Hochberg correction method (adjusted *p* value < 0.05). If at least two of the five metrics suggested significant co-abundance between two OTUs, the relationship was kept in the final network to be represented as an edge. The final co-occurrence network model was displayed by the igraph package in R by using the implementation of the Louvain algorithm to identify communities within each network so that the modularity score of each OTU was maximized within a given network [38].

### 2.9. Quantification and Statistical Analysis

All statistical analyses are reported as the mean ± SD, and the differences in relative abundance of bacterial populations among feces were analyzed using the Mann–Whitney sum rank tests in R software. Significance was declared at *p* < 0.05. All graphs were prepared with R software.

### 2.10. Ethics Approval and Consent of Participants

The study subjects were recruited from the Clinical Trial Center for Functional Foods (CTCF2) in the Chonbuk National University Hospital. Written consent was obtained from all participants. The study was conducted according to the Declaration of Helsinki [39]. The research protocol was approved by the Institutional Review Board (IRB) of Chonbuk National University Hospital, Republic of Korea (CHU_KOREAN_FOOD_2-2_2010).

## 3. Results

### 3.1. Exercise Modified the Composition of the Gut Microbiome More Significantly than Diet Shift

To investigate the role of host factors and diets in the composition of the gut microbiome, we first recruited 30~50-year-old volunteers depending on a meat-containing diet. The 75 volunteers were divided into three groups: one group shifting their diet from a meat diet to a vegetarian diet (the DS group), the second group adopting a 30 min physical exercise in the form of a guided aerobic exercise in a fitness center three times per week without changing their original diets (the EX group), and the control continuing their lifestyle (the Ctrl group). The fecal samples from each group were collected for metagenome analysis by the 16S rRNA sequencing method (DS group, *n* = 14; EX group, *n* = 13; Ctrl group *n* = 14). Sequencing of the V3-V4 sites of the 16Sr rRNA genes of each GI content and feces in each group generated 1137 OTUs by matching with the EzTaxon-e database (http://eztaxon-e.ezbiocloud.net/ Accessed on 10 July 2021) after removal of low-quality sequences or chimeras.

The taxonomically classified OTUs at the phylum level visualized grossly that the gut microbiome was modified by both exercise and diet shift. A maximum-likelihood phylogenetic tree comprising all of the taxa showed that DS increased the abundance of Actinobacteria and decreased the abundance of Bacteroidetes, while EX increased the abundance of Firmicutes and decreased Actinobacteria (Figure 1A,B; see also Appendix A).

The statistical analysis of the mean species diversity by using α-diversity measurements validated that both EX and DS affected the gut microbiome. The α-diversity measurements by the InvSimpson, Shannon, Simpson, and Evenness methods indicated that both EX and DS affected the microbiome, causing it to become more diverse, except for the Evenness index for DS (Figure 1C). Interestingly, all of the α-diversity indices showed that EX affected the gut microbiome more significantly than DS (Figure 1C), meaning that host factors affected the gut microbiome more significantly than diet in determining the composition of the gut microbiome.

### 3.2. Exercise and Diet Shift Modified the Gut Microbiome in Two Different Directions

Since both EX and DS affected the composition of the gut microbiome by increasing its diversity, as shown in Figure 1C, an important question would be the direction of modification by exercise and diet shift. The gross visualization of all of the normalized OTUs at the species level is shown as a heatmap based on the Bray–Curtis distance matrix in Figure 2A. As shown in Figure 2A, the compositions of both gut microbiomes of EX and DS were not only different from each other but also from the control, meaning that EX and DS modified the gut microbiome in two different directions. The hierarchical clustering analysis showed that the gut microbiome of DS was more closely related to the control than EX, although the gut microbiomes of both groups changed. This result is in good agreement with the finding that EX modified the composition of the gut microbiome more significantly than the diet shift, as shown in Figure 1.

A nonmetric multidimensional scaling (NMDS) ordination plot further validated that the gut microbiome compositions of the three groups were quite different from each other (Figure 2B). In accordance with the hierarchical clustering analysis result (Figure 2A), the NMDS ordination plot showed that the gut microbiome of DS was more closely related to the control than EX. Principal coordinate analysis (PCoA) based on the unweighted UniFrac metric also generated similar results (Figure 2C,D). The PCoA plot of Figure 2C shows that the gut microbiomes of the three groups were different from each other, although the gut microbiome of DS was more closely related to the control than EX (ADONIS *p* value 0.013). Measurement of the distance of the centroid on the PCoA plot further validated that the composition of the gut microbiome was modified more significantly by EX than DS (Figure 2D).

### 3.3. Co-Occurrence Network Analysis Showed That Exercise Gave Stronger Selective Pressure to the Gut Microbiome than Diet Shift

To explore the direction and degree of the change in the intestinal microbes constituting gut microbiomes, a bacterial community network analysis was performed for each group (Figure 3). All five *p* values for each method (Spearman and Pearson correlation coefficients, the Mutual Information Score, and the Bray–Curtis and Kullback–Leibler Dissimilarity) were calculated and corrected separately. If at least two of the five metrics’ adjusted *p* value suggested significant (*p**.adj* < 0.05) co-abundance between two OTUs relationship, then co-abundance was considered a strong connection. Only strong connections between OTUs appearing in more than three samples were investigated. The number of nodes and edges increased by both EX and DS (Figure 3). The indices of the community networks (Appendix A) between each group were quite similar, except indices related to the grouping of the OTUs constituting the gut microbiome, such as nodes, edges, and modules.

The total numbers of OTUs present in the gut microbiome of each experimental group were *n* = 793, *n* = 705, and *n* = 847 in Ctrl, EX, and DS, respectively. Although the total numbers of OTUs in each group were similar, the numbers of OTUs connected with other OTUs by a relationship (nodes) were increased in both the gut microbiomes of EX and DS. Because nodes were connected more with each other in DS and EX, the modules in DS and EX were decreased: 28 in control, 8 in EX, and 20 in DS (Figure 3; see also Appendix A). This result suggests that similar kinds of intestinal microbes were increased and unrelated kinds were diminished at the same time in EX and DS, which means that there were selective pressures in both EX and DS to lead the composition of the gut microbiome in a certain direction.

### 3.4. The Abundance of Dialister Succinatiphilus Was Upregulated by Exercise, and the Abundances of Bacteroides Fragilis, Phascolarctobacterium Faecium, and Megasphaera Elsdenii Were Downregulated by Both Exercise and Diet Shift

Since all of the OTUs in this work were classified into nine phyla, we explored the change in the relative abundance of the phyla by DESeq2 [40]. Unsupervised hierarchical clustering of the nine phyla by using DESeq2 showed that EX upregulated the abundances of Tenericutes and Verrucomicrobia and decreased the abundances of Proteobacteria and Lentishaerae (Figure 4; see also Appendix A). DS did not affect the composition of the gut microbiome as much as EX and was only moderately affected, so the decrease in Proteobacteria was not meaningful (Figure 4; see also Appendix A). The dramatic change in the gut microbiome by EX became more evident when comparing EX to DS. It was obvious that EX upregulated the abundances of Tenericutes, Verrucomicrobia, and Acidobacteria, while the abundance of Lentishaerae was decreased (Figure 4; see also Appendix A).

Although more serious disturbance of the gut microbiome by EX than DS could be distinguished at the family level (see also Appendix A), the difference was more evident at the species level (Figure 5, Figure 6 and Figure 7). Most of the species were unknown species in the figures. However, downregulation of the abundances of *Bacteroides fragilis*, *Phascolarctobacterium faecium*, and *Megasphaera elsdenii* and upregulation of *Dialister succinatiphilus* were noticed by EX (Figure 5; Appendix A). Interestingly, the abundances of *Bacteroides fragilis*, *Phascolarctobacterium faecium*, and *Megasphaera elsdenii* were also downregulated by DS, as in the case of EX (Figure 6; see also Appendix A). All of the upregulated bacteria were not taxonomically classified and were unknown bacteria in the DS group. The comparison of EX to DS showed that downregulation of *Veillonella dispar* and upregulation of *Dialister succinatiphilus* were the most noticeable (Figure 7; see also Appendix A.

## 4. Discussion

### 4.1. Host Factors Are More Important than Diet in Determining the Composition of the Gut Microbiome

It has been well proven that both a vegetarian diet and exercise are beneficial to human health. However, the degree of health-beneficial effects and direction by exercise and vegetarian diet are different. Comparative studies have shown that exercise is much more effective in weight loss, reducing the risk of chronic diseases, inducing relaxation and stress relief, and leading to the gain of muscle and bone than a simple vegetarian diet [41,42]. In accordance with comparative studies, this work showed that exercise affected the gut microbiome much more significantly than a vegetarian diet, indicating that host factors are more important than diet in determining the composition of the gut microbiome.

Co-occurrence network analysis further validated the significance of host factors in determining the gut microbiome. In contrast to the increase in microbial diversity by EX (Figure 1C), modules in the co-occurrence network analysis were dramatically decreased from 28 in the control to eight in EX (Figure 3; see also Appendix A). The decrease in modules despite the increase in microbial diversity is because microbial species (OTUs or nodes) were well-connected to each other to be grouped as modules. DS also led to a decrease in modules from 28 to 20. This co-occurrence network analysis indicates that DS posed selective pressure to the gut microbiome, although not as significantly as EX.

This work does not simply emphasize the significance of exercise but rather gives an answer to a fundamental question on how the composition of the gut microbiome is determined. Intestinal microbes obtain their nutrients from the diet of the host. Considering that nutrients are the most important factors for the growth of microbial organisms, the gut microbiome has to be more dependent on diet than host factors if it simply resides in the gut. Surprisingly, this work showed that host factors played a more significant role in determining the composition of the gut microbiome than the diet. The more profound effect on the gut microbiome by exercise than diet shift suggests that the nurturing effect of the gut microbiome by the host for its own purpose plays the main role in determining the composition of the gut microbiome. Therefore, this work suggests that the host nurtures the gut microbiome for its purpose rather than the gut microbiome to drive its host in a certain direction.

### 4.2. Exercise Increased the Abundance of Beneficial Bacteria While Decreasing Harmful Bacteria

The abundances of Tenericutes and Verrucomicrobia were increased by both exercise and a vegetarian diet (Figure 4). However, the phyla were much more dramatically increased by EX and DS. Tenericutes are a group of bacteria without a cell wall and are typically commensals of eukaryotic hosts. Verrucomicrobia are a group of bacteria with compartmentalized cellular structures similar to eukaryotic cells and are frequently found in human feces [43]. Although the abundances of both phyla increased in EX and DS, the abundance of the two phyla was more dramatically increased by EX. Other than Tenericutes and Verrucomicrobia, the abundances of Elusimicrobia and Acidobacteria were increased in EX but not in DS. Overall, the tendency of modification of the gut microbiome in this work was in good agreement with the fact that exercise affects human health more significantly than a vegetarian diet, although both are beneficial [41,42].

Reductions *in Bacteroides fragilis*, *Phascolarctobacterium faecium*, and *Megasphaera elsdenii* were commonly observed in EX and DS at the species level. The deleterious effect of *B. fragilis* is well known. *B. fragilis* is an obligate anaerobe working as an etiological agent of endogenous infections by using its carbohydrate capsule and secretive enzymes [44]. *B. fragilis* is also associated with diarrhea in humans and young farm animals [45,46] and colorectal cancer [47]. Unlike *B. fragilis*, the deleterious role of *P. faecium* and *M.*
*elsdenii* has not been reported. Bacteria are members of the human gut microbiome [48]. *P. faecium* has the ability to use succinate [49], while *M. elsdenii* has the ability to use lactate [50]. Interestingly, all of the bacteria whose abundances were upregulated by EX or DS were unidentified bacteria, except *Dialister succinatiphilus* in EX. *D. succinatiphilus* is a non-spore-forming, Gram-negative bacterium [51]. Although the role of *D. succinatiphilus* in the gut is largely unknown, a fecal transplant experiment on patients showed that increased abundance of *D. succinatiphilus* correlates with the treatment of Tourette syndrome [52]. The clinical study suggests a beneficial role of *D. succinatiphilus* in humans.

### 4.3. The Significance of Host Factors in Determining the Gut Microbiome Is Well-Matched to Evolutionary Evidence That the Composition of the Gut Microbiome Is Determined by the Nurturing Effect of the Host

A recent study on mammalian evolution showed that the composition of the gut microbiome is determined by the nurturing effect of the host [20]. Although all mammals have diverged from a single ancestor, the gut microbiomes of mammals are very different depending on their diets. The prevailing contemporary opinion is that the dietary transitions within an evolutionary lineage determined the diversities of the gut microbiome of each mammal [53,54]. However, Nishida and Ochman showed that the compositions of mammalian gut microbiomes were mainly determined by the physiological changes of the species during the evolutionary process rather than diet shift [20]. In accordance with evolutionary evidence, this work validated that the host nurtures the gut microbiome for its purpose and that host factors very strongly control the gut microbiome.

## 5. Conclusions

Our comparative study showed that host factor modification by exercise affected the gut microbiome more significantly than diet shift, which means that the composition of the gut microbiome is mainly determined by host factors. This work solidifies the recent evolutionary evidence that hosts nurture their own specific gut microbiome so that the diversity of the gut microbiome is mainly determined by host factors rather than diet.

## Figures and Tables

**Figure 1 microorganisms-09-02520-f001:**
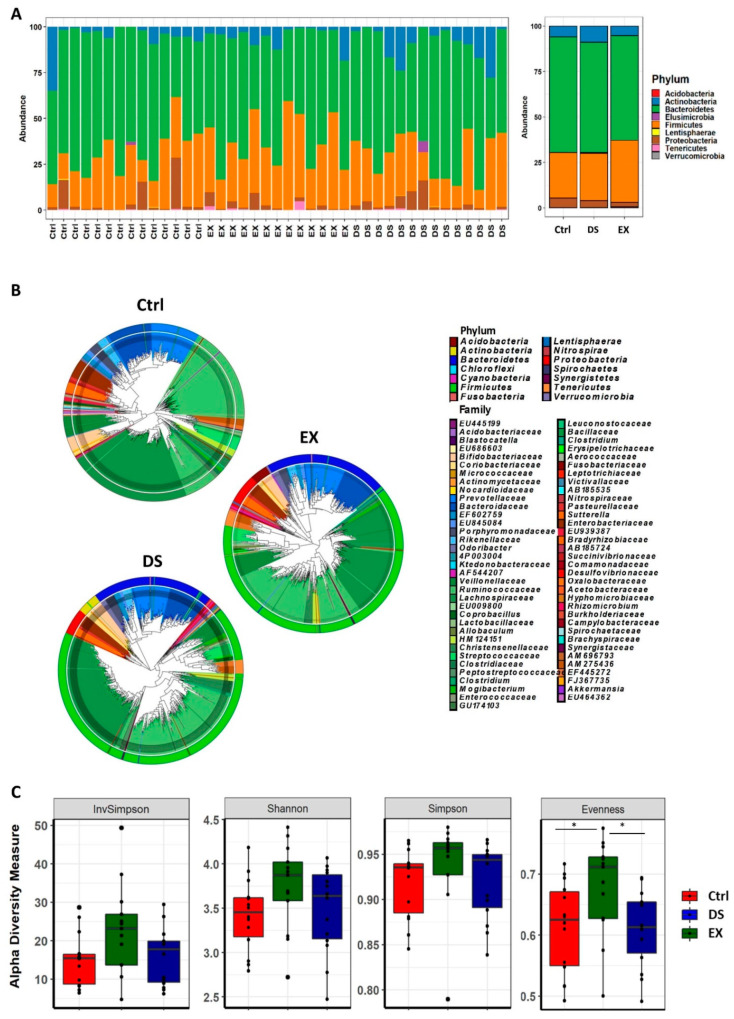
Changes in the composition of the gut microbiome at the phylum level after diet shift or exercise. (**A**) The relative compositional changes in the gut microbiome at the phylum level. (**B**) Maximum-likelihood phylogenetic tree comprising all of the taxa of the gut microbiome in the Ctrl, EX, and DS groups. The rings of the circular dendrogram represent the phylum level, and the corresponding family is depicted in the inner layer. (**C**) α-diversity indexes of the gut microbiome in the Ctrl, EX, and DS groups. α-Diversity values are indicated as the median ± standard deviation. * *p* value < 0.05 was considered as significant. Ctrl, EX, and DS represent the control, exercise, and diet shift groups, respectively.

**Figure 2 microorganisms-09-02520-f002:**
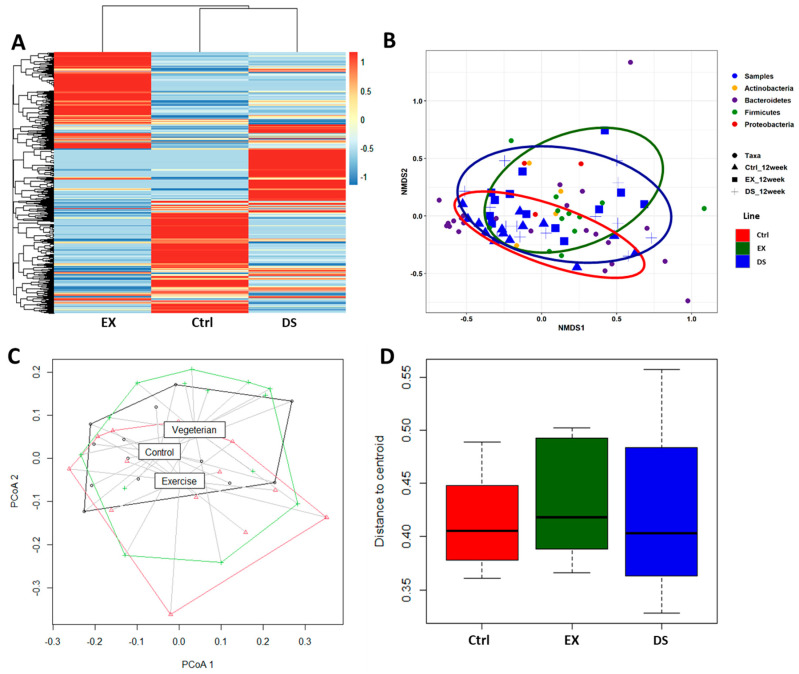
β-diversity comparison of the gut microbiome of the Ctrl, EX, and DS groups. (**A**) Heatmap of the microbial composition for the Ctrl, EX, and DS groups based on the Bray–Curtis distance matrix calculated from normalized OTU values at the species level. (**B**) Nonmetric multidimensional scaling (NMDS) plots showing the difference in the gut microbiome in the Ctrl, EX, and DS groups based on Bray–Curtis distances by using OTUs. (**C**) Principal coordinate analysis (PCoA) based on the unweighted UniFrac metric of the gut microbiome in the Ctrl, EX, and DS groups. (**D**) Distance of centroid for the Ctrl, EX, and DS groups. The Ctrl, EX, and DS represent the control, exercise, and diet shift groups, respectively.

**Figure 3 microorganisms-09-02520-f003:**
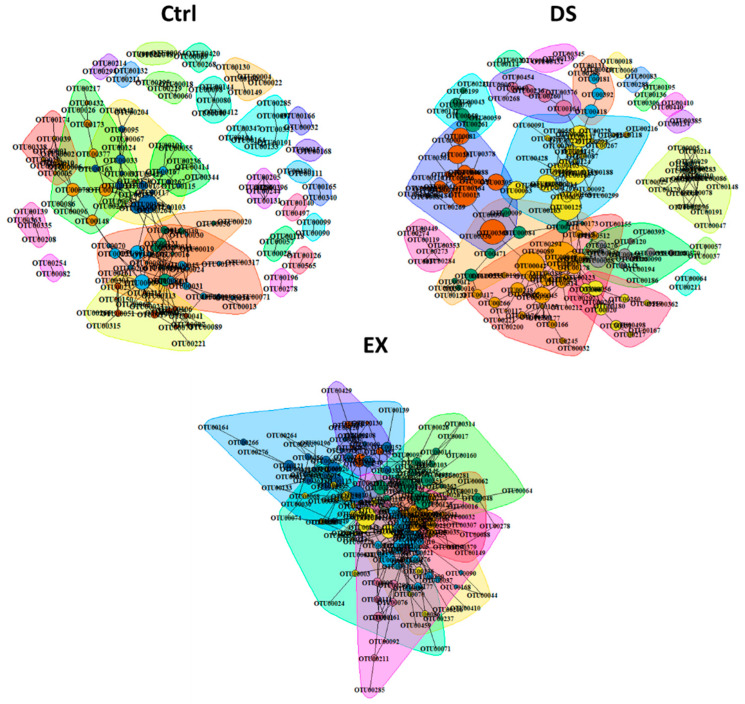
Co-occurrence network analysis by the ReBoot algorithm for the Ctrl, EX, and DS groups. Color-coded network graphs represent the co-occurrence and mutual exclusion interactions among OTUs. White numbers within nodes correspond to numbering in the legend. Transparent shapes represent network communities determined by the Louvain modularity algorithm. Black numbering corresponds to the numbering given to distinguish communities within each network. The Ctrl, EX, and DS represent the control, exercise, and diet shift groups, respectively.

**Figure 4 microorganisms-09-02520-f004:**
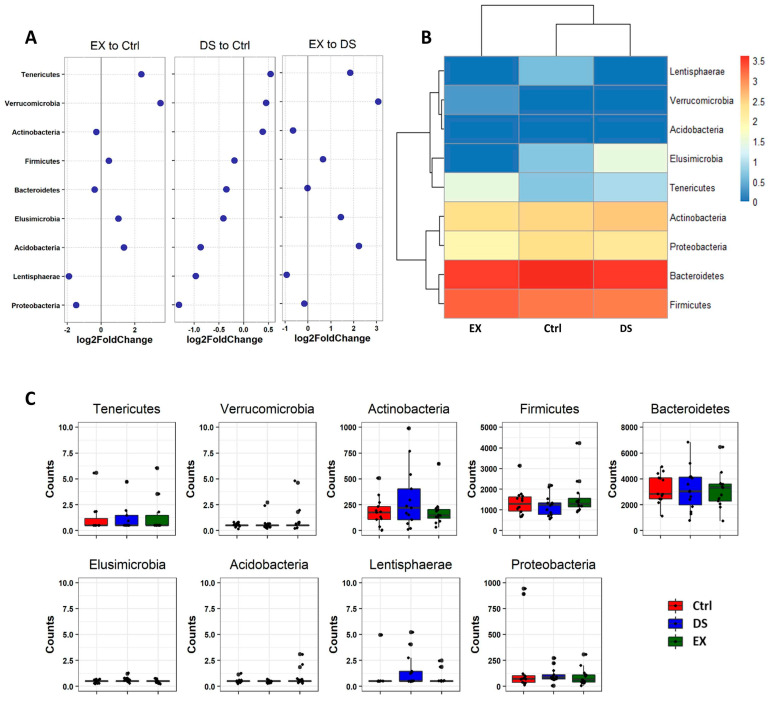
Differential abundance analysis of phylum changes between EX and Ctrl, DS and Ctrl, and EX and DS. (**A**) Log2-fold change in abundance of the phyla constituting the gut microbiome of three experimental groups analyzed by DESeq2 differential abundance analysis. (**B**) Heatmap of nine phyla constituting the gut microbiome of three experimental groups. (**C**) The normalized abundances of nine phyla identified by differential abundance analyses. Boxplots represent normalized count abundances of individual phyla in each group. Ctrl, EX, and DS represent the control, exercise, and diet shift groups, respectively.

**Figure 5 microorganisms-09-02520-f005:**
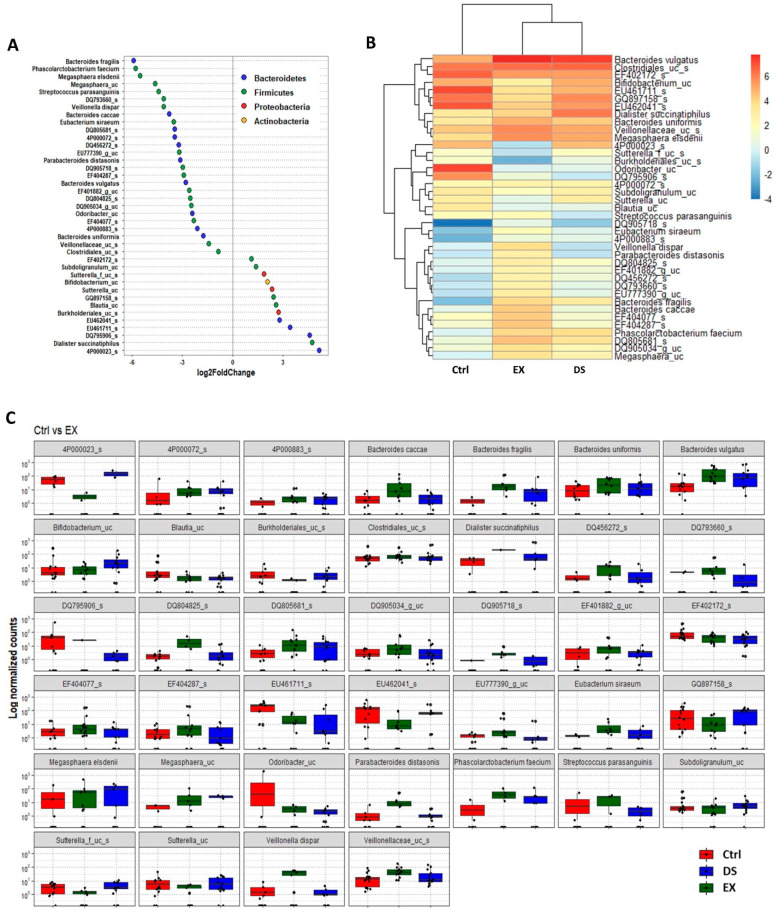
The key taxa changes between Ctrl and EX by differential abundance analysis. (**A**) Log2-fold change in abundance of most abundantly present species in the gut microbiome of the Ctrl and EX groups analyzed by DESeq2 differential abundance analysis. Each point represents a species comparison between two experimental groups. (**B**) Heatmap of most abundantly present species in the Ctrl and EX groups. (**C**) Normalized abundances of 39 significantly different bacterial species of interest that were identified from differential abundance analyses. Boxplots represent normalized count abundances of individual species in each group. *p* value < 0.05 was considered as significant. Ctrl and EX represent the control and exercise groups, respectively.

**Figure 6 microorganisms-09-02520-f006:**
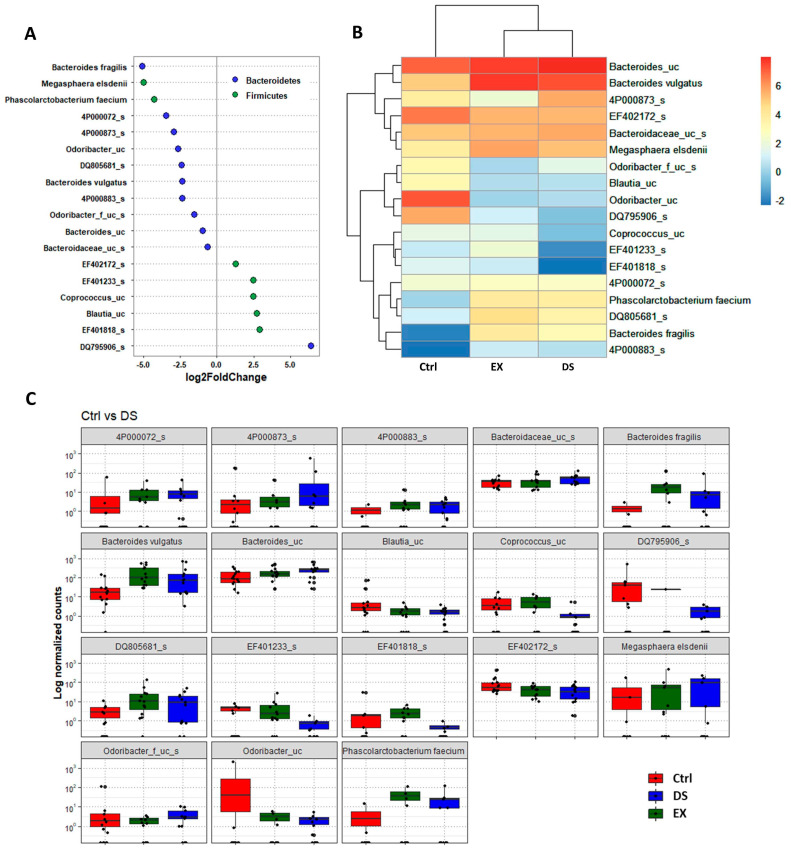
The key taxa changes between Ctrl and DS by differential abundance analysis. (**A**) Log2-fold change in abundance of most abundantly present species in the gut microbiome of the Ctrl and DS groups analyzed by DESeq2 differential abundance analysis. Each point represents a species comparison between two experimental groups. (**B**) Heatmap of most abundantly present species in the Ctrl and DS groups. (**C**) Normalized abundances of 18 significantly different bacterial species of interest that were identified from differential abundance analyses. Boxplots represent normalized count abundances of individual species in each group. *p* value < 0.05 was considered as significant. Ctrl and DS represent the control and exercise groups, respectively.

**Figure 7 microorganisms-09-02520-f007:**
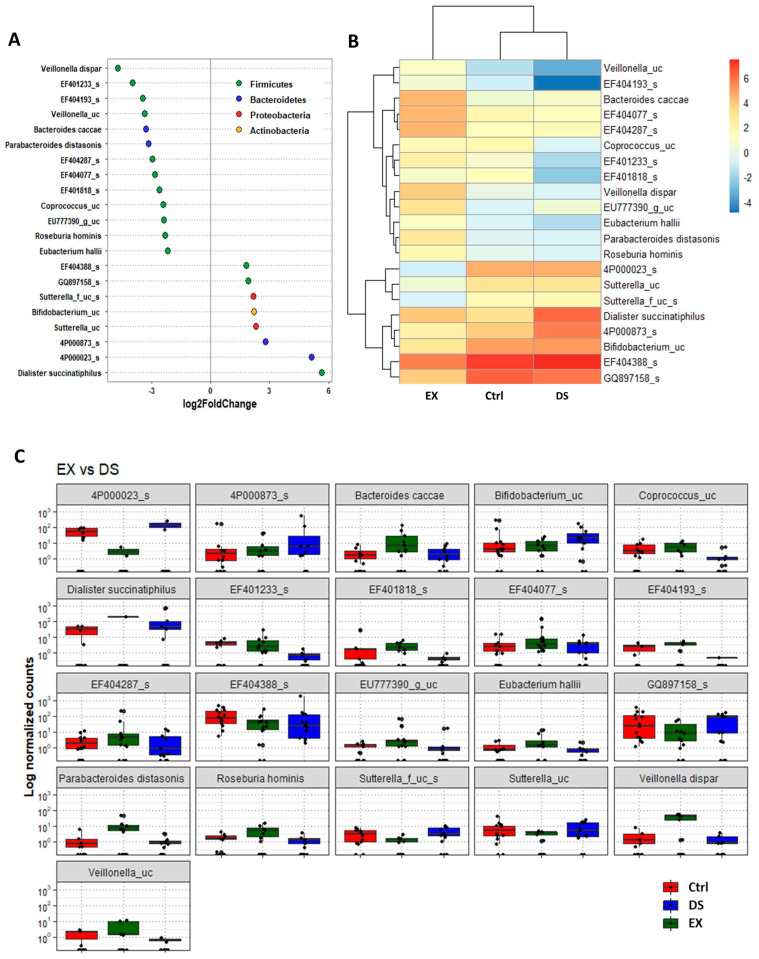
The key taxa changes between EX and DS by differential abundance analysis. (**A**) Log2-fold change in abundance of most abundantly present species in the gut microbiome of the EX and DS groups analyzed by DESeq2 differential abundance analysis. Each point represents a species comparison between two experimental groups. (**B**) Heatmap of most abundantly present species in the EX and DS groups. (**C**) Normalized abundances of 18 significantly different bacterial species of interest that were identified from differential abundance analyses. Boxplots represent normalized count abundances of individual species in each group. *p* value < 0.05 was considered significant. The EX and DS represent the control and exercise groups, respectively.

## Data Availability

The raw data were deposited in the repository at figshare (https://doi.org/10.6084/m9.figshare.16620349.v1).

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
