# Peer review of "Host Factors Affect the Gut Microbiome More Significantly than Diet Shift"

_microorganisms, 2021, doi:10.3390/microorganisms9122520_

Round 1
Reviewer 1 Report
The study aimed to identify the effect of dietary shift versus physical exercise on the gut microbiota. I only have minor comments:
1) Please add references to the first paragraph of the 'introduction'.
2) Line 104, did you mean : at the beginning of the ''study''?
3) lines 12-156 are in red colour.
4) line 264 :A nonmetric: fix the font size.
5) How was the power calculations done?
Author Response
Comments 1: Please add references to the first paragraph of the 'introduction'.
Response 1:
According to your suggestion, we added references to the first paragraph of the introduction. Thanks again for your comment.
Comments 2: Line 104, did you mean: at the beginning of the ''study''?
Response 2:
Yes, you are correct. We really appreciate this comment. Thank you very much. We corrected the mistake. Please refer line 101 for the change.
Comments 3: lines 12-156 are in red colour.
Response 3:
According to your suggestion, we corrected black color.
Comments 4: line 264: A nonmetric: fix the font size.
Response 4:
According to your suggestion, we fixed the font size. Thanks again for your comment.
Comments 5: How was the power calculations done?
Response 5:
This work was to investigate a comparative analysis on the gut microbiome between host factors and diet shift in which null hypothesis was not required. Therefore, we did not calculate the power. Thanks again for your comment.
Thank you very much.
We really appreciate your comments.
Sincerely,
Reviewer 2 Report
The resubmitted manuscript "Host factors affect the gut microbiome more significantly than diet shift" has been review. Some concerns in last round still not well addressed. Below is some to suggest:
Line 80-82: In the section of Introduction, conclusions should not be listed. Instead, the hypothesis should be provided.
I would like to suggest adding the section of Conclusions to further highlight the findings and significance of the current work.
Author Response
Comments 1: Line 80-82: In the section of Introduction, conclusions should not be listed. Instead, the hypothesis should be provided.
Response 1:
According to your suggestion, we re-wrote the portion. Please refer line 77-79 for the change. Thank you very much. It was a great comment.
Comments 2: I would like to suggest adding the section of Conclusions to further highlight the findings and significance of the current work.
Response 2:
According to your suggestion, we added a conclusion section. Please refer line 435-440 for the change. Thank you very much.
Thank you very much.
We really appreciate your comments.
Sincerely,
This manuscript is a resubmission of an earlier submission. The following is a list of the peer review reports and author responses from that submission.
Round 1
Reviewer 1 Report
The research article “The host factor affects the gut microbiome more significantly than diet shift” has been reviewed. The current work aimed to compare the effect of diet shift and physical exercise on the composition of the gut microbiome, and authors found that physical exercise affected gut microbiome more significantly than diet shift. Despite tremendous data it seemingly presented in text and attachment, the effective information it conveys seems limited. The imperfection of sections Abstract and Materials and Methods make me hard to grasp your ideas and innovation. Some concerns require revisions before further evaluation. Below are my detailed concerns:
L1-L2: The title seems intricate, which host factor? More significantly than?
L11: The section of Abstract seems too simple that it did not tell readers how the experiment performed, what exact result it has obtained, and what significances of the current results brings.
L14: The α-diversity analyses by, changed to The α-diversity analyzed by, or The α-diversity analyses of.
L16: The b-diversity analysis, should it β-diversity/beta-diversity? This is a basic information and never should be vague in such an important section like “Abstract”!
L22-23: Please arrange these key words in alphabetical order. Besides, the current key words do not show the content and innovation of this paper well, please once again, concise.
L27: Microbial organisms are no exception, seems monotonous here, try to have a more logical expression.
L34-35: The gut microbiome has been co-evolving with humans throughout its evolutionary history [1-7]. I could not understand why this sentence should be so many references to support.
L35-37: the gut microbiome plays significant determinant roles in almost all phenotypes of animals including diseases as much as the genomes of their hosts [1,2,5,6,7,8,9,10]. What do you want to express by saying “as much as the genomes of their hosts”? Why not cite as [1-2,5-10] a simple way?
L39-42: This sentence looks so long and beyond understanding, please separate it to more sentences to make it easy to readers.
L75-L79: I would like to suggest deleting these sentence, instead, put forward the hypothesis of the current work.
L90: one group shifting their diet from meat diets to a vegetarian diet (the DS group). Did the DS group have an adaptation period from meat-containing diet to vegetarian diet?
L99-103: These sentences were the same as part of 2.10, please attention.
L106-107: How many sample have you collected in total? Have you collected the fecal samples in other period except for week 0 and week 12? These issues must be clearly stated.
L114: How many samples were taken for DNA extraction?
L117-118: DNA concentrations and purity were determined as stated above, where is above? How above says?
L120: Why a commercial company should be cited with a reference? What did you want to express?
L121: The metagenome sequencing and basic analysis were described by Chun et al. [24]. I have noted that [24] was published 2010, does the metagenome sequencing and basic analysis stay the same as over 10 years ago?
L128: Mothur program should be cited as their website suggested.
L136: All raw data involved in this study should be deposited on a public plat, such as NCBI, to share with readers.
L149-151: Unclassified phyla were removed from total samples, and any taxa with a total of less than 0.5% were collected into “other”. Why unclassified phyla should be removed from total samples? Did you mean relative abundance < 0.5% were collected into “other”? If it is true, then why Table S1 in non-published showed < 0.5%? Most often, low abundance does not mean with insignificance.
L155-156: Why should here use QIIME 2, with “A cut-off value of 97% similarity of the 16S rRNA gene sequences was defined as the same species” in L128-129? QIIME2 should cite a reference and define QIIME, NMDS for the first time it appears! Please check similar style throughout the text.
L178-179: Non-normalized abundance data was uploaded to CoNet (39), why here () instead of []?
L209: I cannot oppose your style with conclusion for each result expression, however, it looks really absonant with so many words in 3.4 (L332-334). Remember this is the part of result, just state a fact without any assessment. This is my personal suggestion, anyway.
L357: In the section of Discussion, subheading will make the discussion easier to take for most of readers.
L414: Supplementary Materials, I have noted that the materials you provide is named “non-published”, not supplementary materials, please confirm them as supplementary materials to have more comprehensive cognition of your work.
L442: The reference format is not uniform at all, for example, journal full name or abbreviation, title with capitalizing the first letter or not, page full or abbreviation of the same numbers, etc.
For non-published materials: Why the tables presented as picture without editing?
Author Response
Reviewer 1
Comments and Suggestions for Authors:
The research article “The host factor affects the gut microbiome more significantly than diet shift” has been reviewed. The current work aimed to compare the effect of diet shift and physical exercise on the composition of the gut microbiome, and authors found that physical exercise affected gut microbiome more significantly than diet shift. Despite tremendous data it seemingly presented in text and attachment, the effective information it conveys seems limited. The imperfection of sections Abstract and Materials and Methods make me hard to grasp your ideas and innovation. Some concerns require revisions before further evaluation. Below are my detailed concerns:
L1-L2: The title seems intricate, which host factor? More significantly than?
Response: According to your comment, we modified the tile to clarify meaning. Thank you.
L11: The section of Abstract seems too simple that it did not tell readers how the experiment performed, what exact result it has obtained, and what significances of the current results brings.
Response: According to your comment, we rewrite the abstract. Thank you.
L14: The α-diversity analyses by, changed to The α-diversity analyzed by, or The α-diversity analyses of.
Response: Following your suggestion, we corrected our writing. Please refer line 15 for the change.
L16: The b -diversity analysis, should it β-diversity/beta-diversity? This is a basic information and never should be vague in such an important section like “Abstract”!
Response: We are very sorry for our mistake. It was just a mistake. We corrected it. Thank you very much for your accurate reviewing. Please refer line 17 for the change.
L22-23: Please arrange these key words in alphabetical order. Besides, the current key words do not show the content and innovation of this paper well, please once again, concise.
Response: Following your comment, we rewrote the on your comment,keywords. Please refer line 26 for the change.
L27: Microbial organisms are no exception, seems monotonous here, try to have a more logical expression.
Response: Following your comment, we rewrote on your comment,the sentence. Please refer line 30 for the change.
L34-35: The gut microbiome has been co-evolving with humans throughout its evolutionary history [1-7]. I could not understand why this sentence should be so many references to support.
Response: Following your comment, we reduced the number of citations. Please refer line 38 for the change.
L35-37: the gut microbiome plays significant determinant roles in almost all phenotypes of animals including diseases as much as the genomes of their hosts [1,2,5,6,7,8,9,10]. What do you want to express by saying “as much as the genomes of their hosts”? Why not cite as [1-2,5-10] a simple way?
Response: As your suggested, we corrected it. Thank you. Please refer line 40 for the change.
Thank you again for your suggestion. By “as much as the genomes of their hosts”, we were trying to emphasize the significance of gut microbiome in determining the health and disease of its host.
L39-42: This sentence looks so long and beyond understanding, please separate it to more sentences to make it easy to readers.
Response: As your suggested, we corrected it. Thank you. Please refer line 41~45 for the change.
L75-L79: I would like to suggest deleting these sentences, instead, put forward the hypothesis of the current work.
Response: As your suggested, we corrected it. Thank you. Please refer line 77~80 for the change.
L90: one group shifting their diet from meat diets to a vegetarian diet (the DS group). Did the DS group have an adaptation period from meat-containing diet to vegetarian diet?
Response: There was not any adaptation period. Thank you.
L99-103: These sentences were the same as part of 2.10, please attention.
Response: We deleted the sentence. Thank you.
L106-107: How many sample have you collected in total? Have you collected the fecal samples in other period except for week 0 and week 12? These issues must be clearly stated.
Response: We collected the fecal samples only 2 times: week 0 and week 12. To clarify the state, we changed the sentence as, “Fecal samples were freshly collected 2 times from each participant at the day beginning (week 0) and the end of the intervention (week 12).” Please refer line 100~101 for the change.
L114: How many samples were taken for DNA extraction?
Response: We were taken ~1g per sample for DNA extraction. We modified the sentence to clarify it as, “Genomic DNA was extracted from ~1g fecal aliquots samples…”
Please refer line 108 for the change.
L117-118: DNA concentrations and purity were determined as stated above, where is above? How above says?
Response: Thank you for your conscious comment. We replaced the sentence as “DNA was quantified using a BioSpec-nano spectrophotometer (Shimadzu, Kyoto, Japan)”.
Please refer line 111~112 for the change.
L120: Why a commercial company should be cited with a reference? What did you want to express?
Response: Thank you for your comment. We removed the reference [23].
L121: The metagenome sequencing and basic analysis were described by Chun et al. [24]. I have noted that [24] was published 2010, does the metagenome sequencing and basic analysis stay the same as over 10 years ago?
Response: There is not a problem in removing this sentence since the metagenome sequencing and basic analysis were well explained and basically performed again by our lab for validation. Thanks again for your comment.
L128: Mothur program should be cited as their website suggested.
Response: We did not use the Mothur program. We corrected the sentence as “The number of sequences analyzed, observed diversity richness [Operational Taxonomic Units (OTUs)], estimated OTU richness (ACE and Chao1), and Shannon diversity index indicated in Table S1 were calculated using the phyloseq (1.28.0) package in R version 3.6.1 [26].”
Please refer line 124~125 for the change.
L136: All raw data involved in this study should be deposited on a public plat, such as NCBI, to share with readers.
Response: We are in the process of depositing the raw data on NCBI. Thank you.
L149-151: Unclassified phyla were removed from total samples, and any taxa with a total of less than 0.5% were collected into “other”. Why unclassified phyla should be removed from total samples? Did you mean relative abundance < 0.5% were collected into “other”? If it is true, then why Table S1 in non-published showed < 0.5%? Most often, low abundance does not mean with insignificance.
Response: Thank you for your precise comment. There were some mistakes in the manuscript. We apologize for this inaccurateness. Corrections are as followings:
- As it mentioned in citation [26] we used a prevalence filtering method “to avoid spending much time analyzing taxa that were only rarely seen. This is a useful method to filter noises (taxa that are actually just artifacts during data collection process). We wrote the filtering method more clearly as “In brief, the OTUs that are present as a single unit in each sample were considered as OTUs generated by sequencing errors, thereby removed by prevalence filtering method (threshold=2) for further analysis. Please refer line 131~132 for the change.
- The phyla which have less than 5% relative abundancy were classified into ‘other’ group for visualization of abundance. In case of other data analysis all the OTUs were used. We did not remove the unclassified phyla. We rewrote previous description. Please refer line 140~143 for the change.
- Table S1 in supplementary material showed 100% phylum data from the normalized relative abundance data. All the OTUs were used to calculate phylum level abundance.
- Low abundance does not mean insignificant as you said. We used all the OTUs for data analysis in this work. I am sorry. It seems that we made you misunderstood in our previous improper description.
L155-156: Why should here use QIIME 2, with “A cut-off value of 97% similarity of the 16S rRNA gene sequences was defined as the same species” in L128-129? QIIME2 should cite a reference and define QIIME, NMDS for the first time it appears! Please check similar style throughout the text.
Response: Thank you for your comment. We did not use the QIIME to identify similarity or for other analysis. We clarified the previous writing, and added the related citations as “A cut-off value of 97% similarity of the 16S rRNA gene sequences was defined as the same species which was defined by using the algorithm of Myers & Miller (1988) as previously described[25]”.
Please refer line 121~122 for the change.
We also corrected the previous description as “Unweighted PCoA was calculated and visualized by vegan package [30], while NMDS was plotted in the phyloseq package in R.” Please refer line 147~149 for the change.
L178-179: Non-normalized abundance data was uploaded to CoNet (39), why here () instead of [ ]?
Response: Thanks again for your comment. We changed the () to [].
Please refer line 171 for the change.
L209: I cannot oppose your style with conclusion for each result expression, however, it looks really absonant with so many words in 3.4 (L332-334). Remember this is the part of result, just state a fact without any assessment. This is my personal suggestion, anyway.
Response: According to your suggestion, we moved and rewrote the description in discussion. Please refer line 355~362 for the change.
L357: In the section of Discussion, subheading will make the discussion easier to take for most of readers.
Response: Thanks again for your comment. We added subheading in Discussion as you suggested. Please refer Discussion for the change.
L414: Supplementary Materials, I have noted that the materials you provide is named “non-published”, not supplementary materials, please confirm them as supplementary materials to have more comprehensive cognition of your work.
Response: We corrected the mistake. Thanks again for your comment.
L442: The reference format is not uniform at all, for example, journal full name or abbreviation, title with capitalizing the first letter or not, page full or abbreviation of the same numbers, etc.
Response: We reformatted all references by End Note as MDPI suggested. Please refer line 442~552 for the change.
For non-published materials: Why the tables presented as picture without editing?
Response: As your requested, we have modifiedchanged all tables (supplementary materials).
Reviewer 2 Report
The current study investigated the effect of shifting from meat based diet to either vegetarian diet or physical exercise vs control who didn't change their diet on the gut microbiota.
Introduction:
1) The last paragraph from lines 76-79 belong to conclusion. I suggest changing its place to conclusions and rather elucidating the aim of the study.
Materials and methods:
2) line 96, n=20 , n should be in italic. The total number of participants in each groups is 63, not 75 as previously notes.
3) line 99, please add a reference after Declaration of Helsinki.
4) lines 96-103 are repeated again in 200-207, I suggest you delete lines 96-103.
5) What is the inclusion and exclusion criteria for the study?
Results:
6) lines 210 -218 are repeated. Please revise the text and not repeat information. The total number of participants in line 217 is changed from 64 to 41 which is not 75 volunteers are previously mentioned. Were there any exclusions?
7) line 223: excise should be changed to exercise.
8) Figure 3: what is VT? do you mean DS? please fix it.
9) What kind of exercise did the participants perform? anything specific or just general exercise?
Author Response
Reviewer 2
Comments and Suggestions for Authors
The current study investigated the effect of shifting from meat based diet to either vegetarian diet or physical exercise vs control who didn't change their diet on the gut microbiota.
Introduction:
1) The last paragraph from lines 76-79 belong to conclusion. I suggest changing its place to conclusions and rather elucidating the aim of the study.
Response: As your suggested, we corrected it. Thank you. Please refer line 77~80 for the change.
Materials and methods:
2) line 96, n=20, n should be in italic. The total number of participants in each groups is 63, not 75 as previously notes.
Response: At your request, we changed as italic n. Thank you.
The total number of participants was 41. The volunteers were 75. The remaining 34 participants fell out during the study. To clarify the sentence we changed the sentence as “ ..the fecal samples from 41 individuals who followed the guideline were collected for further analysis (the DS group, n=14; the EX group, n=13; the Ctrl group n=14).” Please refer line 95~97 for the change.
3) line 99, please add a reference after Declaration of Helsinki.
Response: We add a reference after Declaration of Helsinki. Thank you. Please refer line 195~196 and citation [38] for the change.
4) lines 96-103 are repeated again in 200-207, I suggest you delete lines 96-103.
Response: As your suggested, we deleted lines 96-103. Thank you.
5) What is the inclusion and exclusion criteria for the study?
Response: The volunteers were 75. The total number of participants was 63 (the DS group, n=20; the EX group, n=21; the Ctrl group n=22). After 3 months, the volunteers were interviewed to ask whether they strictly followed the experimental guideline, we excluded the precipitant who did not follow any guidelines.
Results:
6) lines 210 -218 are repeated. Please revise the text and not repeat information. The total number of participants in line 217 is changed from 64 to 41 which is not 75 volunteers are previously mentioned. Were there any exclusions?
Response: We corrected the mistakes. Thank you.
7) line 223: excise should be changed to exercise.
Response: We corrected the mistake. Thank you.
8) Figure 3: what is VT? do you mean DS? please fix it.
Response: VT is DS. We changed the mistake in Figure 3. Thank you.
9) What kind of exercise did the participants perform? anything specific or just general exercise?
Response: The exercise was a 30 min physical exercise of a guided aerobic general exercise in a fitness center three times per week. Thank you.
Reviewer 3 Report
In this work, the authors have compared the effect of diet shift (DS) from heavily relying on meatatarian diets to vegetarian diets and physical exercise (EX) on the composition of the gut microbiome. However, some aspects should be improved before the publication. Introduction should be improve adding the references regarding the role of physical activity and diet on gut microbiota (e.g. Dorelli et al. Can Physical Activity Influence Human Gut Microbiota Composition Independently of Diet? A Systematic Review. Nutrients. 2021 May 31;13(6):1890. doi: 10.3390/nu13061890). In Material and methods, the study design should be better clarified perhaps by using a flow chart to make it clearer to the reader. Moreover, the author should add the criteria of exclusion or inclusion of subjects. How did you determine the subgroups?
In figure 3 there seems to have been an error.
Author Response
Reviewer 3
In this work, the authors have compared the effect of diet shift (DS) from heavily relying on meatatarian diets to vegetarian diets and physical exercise (EX) on the composition of the gut microbiome. However, some aspects should be improved before the publication. Introduction should be improve adding the references regarding the role of physical activity and diet on gut microbiota (e.g. Dorelli et al. Can Physical Activity Influence Human Gut Microbiota Composition Independently of Diet? A Systematic Review. Nutrients. 2021 May 31;13(6):1890. doi: 10.3390/nu13061890).
Response:
As your suggested, we add reference [20]. Thank you. Please refer line 484~486 for the change. Thank you.
In Material and methods, the study design should be better clarified perhaps by using a flow chart to make it clearer to the reader. Moreover, the author should add the criteria of exclusion or inclusion of subjects. How did you determine the subgroups?
Response:
The volunteers were 75. The total number of participants was 63 (the DS group, n=20; the EX group, n=21; the Ctrl group n=22). After 3 months, the volunteers were interviewed to ask whether they strictly followed the experimental guideline, we excluded the precipitant who did not follow any guidelines. The remaining 34 participants fell out during the study. To clarify the sentence, we changed the sentence as “the fecal samples from 41 individuals who followed the guideline were collected for further analysis (the DS group, n=14; the EX group, n=13; the Ctrl group n=14).” Please refer line 95~97 for the change.
As your suggested, we add flow chart (supplementary Figure S1). Please refer line 97 for the change. Figure S1. Flow chart for the study subjects. Thank you.
In figure 3 there seems to have been an error.
Response: VT is DS. We changed the mistake in Figure 3. Thank you.
Round 2
Reviewer 1 Report
The revised version of manuscript has been reviewed. It is still far from publication, I know many concerns were not well addressed. Below are my certain concerns:
L1-L2: The title seems intricate, which host factor? More significantly than?
Response: According to your comment, we modified the tile to clarify meaning. Thank you.
Comments: Still not claer.
L11: The section of Abstract seems too simple that it did not tell readers how the experiment performed, what exact result it has obtained, and what significances of the current results brings.
Response: According to your comment, we rewrite the abstract. Thank you.
Comments: Why not use Tracking marks instead of red text?
L22-23: Please arrange these key words in alphabetical order. Besides, the current key words do not show the content and innovation of this paper well, please once again, concise.
Response: Following your comment, we rewrote the on your comment,keywords. Please refer line 26 for the change.
Comments: Still not the way we suggested.
L90: one group shifting their diet from meat diets to a vegetarian diet (the DS group). Did the DS group have an adaptation period from meat-containing diet to vegetarian diet?
Response: There was not any adaptation period. Thank you.
Commments: Did you think it is reasonable to have no adaptation period?
L114: How many samples were taken for DNA extraction?
Response: We were taken ~1g per sample for DNA extraction. We modified the sentence to clarify it as, “Genomic DNA was extracted from ~1g fecal aliquots samples…”
Comments: How many, not how much!
L117-118: DNA concentrations and purity were determined as stated above, where is above? How above says?
Response: Thank you for your conscious comment. We replaced the sentence as “DNA was quantified using a BioSpec-nano spectrophotometer (Shimadzu, Kyoto, Japan)”.
Comments: This is far from what I want to ask!
L128: Mothur program should be cited as their website suggested.
Response: We did not use the Mothur program. We corrected the sentence as “The number of sequences analyzed, observed diversity richness [Operational Taxonomic Units (OTUs)], estimated OTU richness (ACE and Chao1), and Shannon diversity index indicated in Table S1 were calculated using the phyloseq (1.28.0) package in R version 3.6.1 [26].”
Please refer line 124~125 for the change.
Comments: Why not use Mothur program?
L136: All raw data involved in this study should be deposited on a public plat, such as NCBI, to share with readers.
Response: We are in the process of depositing the raw data on NCBI. Thank you.
Comments: This process could be finished in one way at most! Therefore, this is not the exact reason for this question!
L149-151: Unclassified phyla were removed from total samples, and any taxa with a total of less than 0.5% were collected into “other”. Why unclassified phyla should be removed from total samples? Did you mean relative abundance < 0.5% were collected into “other”? If it is true, then why Table S1 in non-published showed < 0.5%? Most often, low abundance does not mean with insignificance.
Response: Thank you for your precise comment. There were some mistakes in the manuscript. We apologize for this inaccurateness. Corrections are as followings:
- As it mentioned in citation [26] we used a prevalence filtering method “to avoid spending much time analyzing taxa that were only rarely seen. This is a useful method to filter noises (taxa that are actually just artifacts during data collection process). We wrote the filtering method more clearly as “In brief, the OTUs that are present as a single unit in each sample were considered as OTUs generated by sequencing errors, thereby removed by prevalence filtering method (threshold=2) for further analysis. Please refer line 131~132 for the change.
- The phyla which have less than 5% relative abundancy were classified into ‘other’ group for visualization of abundance. In case of other data analysis all the OTUs were used. We did not remove the unclassified phyla. We rewrote previous description. Please refer line 140~143 for the change.
- Table S1 in supplementary material showed 100% phylum data from the normalized relative abundance data. All the OTUs were used to calculate phylum level abundance.
- Low abundance does not mean insignificant as you said. We used all the OTUs for data analysis in this work. I am sorry. It seems that we made you misunderstood in our previous improper description.
Comments: Did you have any reasons for the explanation by saying these?
L155-156: Why should here use QIIME 2, with “A cut-off value of 97% similarity of the 16S rRNA gene sequences was defined as the same species” in L128-129? QIIME2 should cite a reference and define QIIME, NMDS for the first time it appears! Please check similar style throughout the text.
Response: Thank you for your comment. We did not use the QIIME to identify similarity or for other analysis. We clarified the previous writing, and added the related citations as “A cut-off value of 97% similarity of the 16S rRNA gene sequences was defined as the same species which was defined by using the algorithm of Myers & Miller (1988) as previously described[25]”.
Comments: this is not the modern way for analyzing these data!
L442: The reference format is not uniform at all, for example, journal full name or abbreviation, title with capitalizing the first letter or not, page full or abbreviation of the same numbers, etc.
Response: We reformatted all references by End Note as MDPI suggested. Please refer line 442~552 for the change.
Comments: Still not the way the journal require! If you did not do like that, than I will reject to review that!
For non-published materials: Why the tables presented as picture without editing?
Response: As your requested, we have modifiedchanged all tables (supplementary materials).
Author Response
L1-L2: The title seems intricate, which host factor? More significantly than?
Response: According to your comment, we modified the tile to clarify meaning. Thank you.
Comments: Still not clear.
Response: We discussed about it and consulted with peers. However, we like to stay in the title. We are trying to present the key concept of our manuscript in the title rather than emphasizing specificity. We would really appreciate it if you understand.
Thank you.
L11: The section of Abstract seems too simple that it did not tell readers how the experiment performed, what exact result it has obtained, and what significances of the current results brings.
Response: According to your comment, we rewrite the abstract. Thank you.
Comments: Why not use Tracking marks instead of red text?
Response: Since we changed significantly the abstract, we marked the change in red.
Here is the original text of the abstract.
“The determining factors of the composition of the gut microbiome are one of the main interests in current science. In this work, we compared the effect of diet shift (DS) from heavily relying on meatatarian diets to vegetarian diets and physical exercise (EX) on the composition of the gut microbiome. The α-diversity analyses by InvSimpson, Shannon, Simpson, and Evenness showed that both EX and DS affected the microbiome to be more diverse, but EX affected gut microbi-ome more significantly than DS. The b-diversity analysis indicated that EX and DS modified the gut microbiome into two different directions. Co-occurrence network analysis confirmed that both EX and DS modified gut microbiome into different directions although EX modified more significantly. Most notably, the abundance of Dialister succinatiphilus was upregulated by EX, and the abundances of Bacteriodes fragilis, Phascolarctobacterium faecium, and Megasphaera elsdenii were downregulated by both EX and DS.”
L22-23: Please arrange these key words in alphabetical order. Besides, the current key words do not show the content and innovation of this paper well, please once again, concise.
Response: Following your comment, we rewrote keywords.
Please refer line 26 for the change.
Comments: Still not the way we suggested.
Response: We modified key words again. Thank you. Please refer line 26 for the change.
L90: one group shifting their diet from meat diets to a vegetarian diet (the DS group). Did the DS group have an adaptation period from meat-containing diet to vegetarian diet?
Response: There was not any adaptation period. Thank you.
Commments: Did you think it is reasonable to have no adaptation period?
Response: We understand your concern. Since we want to see the compositional change after dramatic change of diet, we did not include an adaptation period. Thank you.
L114: How many samples were taken for DNA extraction?
Response: We were taken ~1g per sample for DNA extraction. We modified the sentence to clarify it as, “Genomic DNA was extracted from ~1g fecal aliquots samples…”
Comments: How many, not how much!
Response: We describe the procedure in more detail. Thank you. Please refer line 108-109 for the change.
L117-118: DNA concentrations and purity were determined as stated above, where is above? How above says?
Response: Thank you for your conscious comment. We replaced the sentence as “DNA was quantified using a BioSpec-nano spectrophotometer (Shimadzu, Kyoto, Japan)”.
Comments: This is far from what I want to ask!
Response: We rewrite the procedure. Thank you. Please refer line 113-115 for the change.
L128: Mothur program should be cited as their website suggested.
Response: We did not use the Mothur program. We corrected the sentence as “The number of sequences analyzed, observed diversity richness [Operational Taxonomic Units (OTUs)], estimated OTU richness (ACE and Chao1), and Shannon diversity index indicated in Table S1 were calculated using the phyloseq (1.28.0) package in R version 3.6.1 [26].”
Please refer line 124~125 for the change.
Comments: Why not use Mothur program?
Response: We corrected mistake. We really appreciate your comment. Thank you so much. Please refer line 124~125 for the change.
L136: All raw data involved in this study should be deposited on a public plat, such as NCBI, to share with readers.
Response: We are in the process of depositing the raw data on NCBI. Thank you.
Comments: This process could be finished in one way at most! Therefore, this is not the exact reason for this question!
Response: Following your suggestion, we deposited in Figshare (The raw data deposited in the repository at figshare (https://doi.org/10.6084/m9.figshare.16620349.v1).). Please refer line 129~131.
L149-151: Unclassified phyla were removed from total samples, and any taxa with a total of less than 0.5% were collected into “other”. Why unclassified phyla should be removed from total samples? Did you mean relative abundance < 0.5% were collected into “other”? If it is true, then why Table S1 in non-published showed < 0.5%? Most often, low abundance does not mean with insignificance.
Response: Thank you for your precise comment. There were some mistakes in the manuscript. We apologize for this inaccurateness. Corrections are as followings:
- As it mentioned in citation [26] we used a prevalence filtering method “to avoid spending much time analyzing taxa that were only rarely seen. This is a useful method to filter noises (taxa that are actually just artifacts during data collection process). We wrote the filtering method more clearly as “In brief, the OTUs that are present as a single unit in each sample were considered as OTUs generated by sequencing errors, thereby removed by prevalence filtering method (threshold=2)for further analysis. Please refer line 131~132 for the change.
- The phyla which have less than 5% relative abundancy were classified into ‘other’ group for visualization of abundance. In case of other data analysis all the OTUs were used. We did not remove the unclassified phyla. We rewrote previous description. Please refer line 140~143 for the change.
- Table S1 in supplementary material showed 100% phylum data from the normalized relative abundance data. All the OTUs were used to calculate phylum level abundance.
- Low abundance does not mean insignificant as you said. We used all the OTUs for data analysis in this work. I am sorry. It seems that we made you misunderstood in our previous improper description.
Comments: Did you have any reasons for the explanation by saying these?
Response: There is not a specific reason. We just described the experimental procedures and results.
L155-156: Why should here use QIIME 2, with “A cut-off value of 97% similarity of the 16S rRNA gene sequences was defined as the same species” in L128-129? QIIME2 should cite a reference and define QIIME, NMDS for the first time it appears! Please check similar style throughout the text.
Response: Thank you for your comment. We did not use the QIIME to identify similarity or for other analysis. We clarified the previous writing, and added the related citations as “A cut-off value of 97% similarity of the 16S rRNA gene sequences was defined as the same species which was defined by using the algorithm of Myers & Miller (1988) as previously described [25]”.
Comments: this is not the modern way for analyzing these data!
Response: We corrected the description again reflecting your opinion. Thank you. Please refer line 122~125 for the change.
L442: The reference format is not uniform at all, for example, journal full name or abbreviation, title with capitalizing the first letter or not, page full or abbreviation of the same numbers, etc.
Response: We reformatted all references by End Note as MDPI suggested. Please refer line 442~552 for the change.
Comments: Still not the way the journal require! If you did not do like that, than I will reject to review that!
Response: We corrected our mistake. We are really sorry for the mistake. Please refer line 447~572 for the change.
For non-published materials: Why the tables presented as picture without editing?
Response: As your requested, we have modified changed all tables (supplementary materials).
Response: We confirmed it. Thank you.